# A Zero-Dimensional Mixing Controlled Combustion Model for Real Time Performance Simulation of Marine Two-Stroke Diesel Engines

**Yongming Feng [1,\*], Haiyan Wang [2], Ruifeng Gao [1] and Yuanqing Zhu [1]**

[1] College of Power and Energy Engineering, Harbin Engineering University, Harbin 150001, China; gaoruifeng@hrbeu.edu.cn (R.G.); zhuyuanqing@hrbeu.edu.cn (Y.Z.)
[2] Merchant Marine College, Shanghai Maritime University, Shanghai 201306, China; wanghaiyan@shmtu.edu.cn
[\*] Correspondence: fengyongming@hrbeu.edu.cn; Tel.: +86-451-82568384

**Abstract:** The paper presents a performance prediction model of marine low-speed two-stroke diesel engines based on an advanced MCC (mixture controlled combustion) model coupled with a fuel injection model. Considering the time of real calculation, the so-called "concentrated exhausting gas" scavenging model and the working process model are used in the present work, and improved by introducing the ratio of pure combustion product over the total gas mass in the cylinder as an expression of the working medium components. The reaction rate model in the zero-dimensional MCC model is improved by introducing the fraction of combustion product in the fuel spray, and the relationship between the combustion model and scavenging quality is established. Meanwhile, the combustion model was simplified in the diffusion combustion phases and integrated with the fuel injection model in order to respond to the change of injection profile and injection timing. A large-scale low-speed marine diesel engine was used for a simulation. The results of the whole model are consistent with experimental data and the speed of calculation is fast enough for real time simulation of low speed and medium speed diesel engines. The prediction model can be used in the design and calibration of the electronic control system and performance optimization of the marine two-stroke diesel engine.

**Keywords:** marine two-stroke diesel engine; mixing controlled combustion model; real-time modeling; performance simulation

---

## 1. Introduction

To comply with stricter and stricter regulations on environmental protection, manufacturers of diesel engines developed and are developing new technology to reduce emission of $NO_x$, $CO_2$, and other pollutant emissions. There are a lot of parameters that need to be adjusted or monitored during engine running. These parameters can be classified as three types: inputs that can be modified, parameters that provide information about the real system state, and the engine outputs [1]. The desired value of outputs can be achieved by adjusting inputs which in turn are adjusted by engine control system. However, calibration of the engine and its controller requires extensive activity at the engine test bench. Therefore, a model for the real-time calculation of the engine working cycle could be considered an important aid [2].

The control-oriented model traditionally includes two types of models: one is the mean value engine model, which is based on the engine principle; the other includes the parametric linear model, the "black box" model, etc, which are based on experiments and the system identification method [3].

These two types of models have been used widely for control synthesis and calibration [4–9]. However, these models don't include the combustion model, and cannot predict in-cylinder pressure and other parameters related to the combustion process. Therefore, crank angle resolved engine models with various combustion models are becoming more important in performance prediction and engine control.

The diesel engine model is composed of many different modules, including the intake manifold, exhaust manifold, working process in cylinder, turbocharger, etc. In these modules, the model of working process in the cylinder is critical to predicting the performance of the engine. Based on different computing methods, the engine model can be divided into two types: multi-dimensional and thermodynamic. The thermodynamic model can be further divided into single zone and multi zone [10]. The multi-dimensional model can be solved by using three-dimensional computational fluid dynamics (CFD) software. However it is impractical for the simulation of the working cycle because of its extremely high calculation load. In fact, it needs more than 100 times the real running time even for solving a 1-dimensional model [11]. Moreover, the multi zone thermodynamic model usually requires more than real time [12]. Although there were several papers that show real time multi zone combustion models [2,13], these models cannot be used in real time simulation. The reason is that these models don't couple with working process models and require more time than real time for simulation of the working process. In fact, the computational time required is near 0.5 ms per crank angle on a Pentium D PC [2]. If coupled with the working process, especially the scavenging process, it will require more time. Therefore, the single zone model may be the only choice for real time simulation.

The single zone model is based on the first law of thermodynamics, and the cylinder is taken as a whole, in which gas composition, temperature, and pressure are equal everywhere. This type of model is widely used for the simulation of engines [14–16] and for calculation of the heat release rate of fuel [17–19]. Wiebe's function is one of the widely used combustion models in working process and engine performance simulation [20,21]. Moreover, the parameters of Wiebe's function are optimized by applying experimental data and adjustable corresponding to the changing of the running conditions of the engine [22,23]. However, more and more parameters, such as injection timing, injection profile, multi-injection, and optimization of CA50, etc., of the fuel injection system of the sophisticated engine can be controlled by the ECU (engine control unit). Wiebe's function cannot fully provide these flexibilities. Therefore, various zero-dimension combustion models, most of which usually couple with the emission model of NOx, are applied in many cases [12,23,24].

Considering the objective of the present work is to put forward a real time model, a kind of heat release rate model is followed. Chmela [25] put forward a MCC (mixture controlled combustion) model, in which the reaction rate of fuel is controlled by the density of turbulent kinetic energy. Chmela [26] put forward another model based on the three combustion phases, in which reaction rate is controlled by a Magnussen type equation and Arrhenius type equation. Rezaei [27] combined spray model with a heat release model and adopted a different laminar time scale equation. Katrasnik [28,29] put forward an innovative mechanistically based spray model intended for use in the MCC model of a four stroke engine, and split the combustion into three phases by splitting the diffusion combustion into a rich combustion (in the centre of the spray) and a lean combustion (edges of the spray). Dowell et al [30] further extended the MCC model to EGR (exhaust gas recirculation) engine simulation. These models can accurately predict the heat release rate of diesel engines and can be used for real-time performance simulation of diesel engines. However, there are still some problems to be solved, especially for marine two-stroke diesel engines.

Firstly, the fuel system of a modern engine usually has the function of multi-injection, but the existing models don't show how to deal with this kind of function. If a multi-injection or post-injection occurs, does the model mode change into the former stage of combustion from the later? Secondly, if the gas exchanging process is very important to the performance of the marine two-stroke engine, then how does the combustion model response if the quality of the gas exchanging varies? The objective of the present work is initially to find a solution for these two problems. The aim of this paper is to create a crank angle resolved, real time capable, performance prediction model of marine low-speed

two-stroke diesel engines based on the zero-dimensional MCC model, which can closely combine the fuel injection law with the working process of diesel engines, so that the diesel engine performance simulation model can respond to the changes of the fuel injection law and scavenging quality, and has the ability of real-time simulation.

## 2. Model of Working Cycle

The single zone working cycle model is based on the first law of thermodynamics (Equation (1)), mass conservation equation (Equation (2)), and ideal gas equation (Equation (3)), shown as follows:

$$\frac{d(MzUz)}{dt} = \frac{dQ_f}{dt} + m_s h_s - m_e h_e - \frac{dQ_w}{dt} - p_z \frac{dV_z}{dt}, \tag{1}$$

$$\frac{dm_z}{dt} = \frac{1}{LHV} \frac{dQ_f}{dt} + m_s - m_e, \tag{2}$$

$$p_z V_z = m_z R_z T_z. \tag{3}$$

Equation (1) can be extended and rearranged to obtain Equation (4). Therefore, the gas temperature in the cylinder is taken as the initial state variable of the calculation.

$$\frac{dT_z}{dt} = \frac{1}{m_z c_{vz}} \left( \frac{dQ_f}{dt} + m_s h_s - m_e h_e - \frac{dQ_w}{dt} - p_z \frac{dV_z}{dt} - c_{vz} T_z \frac{dM_z}{dt} \right), \tag{4}$$

where, $d(M_z U_z)/dt$ is the rate of the change of internal energy; $dQ_f/dt$ is the heat release rate of fuel; $m_s$ is the mass flow of the scavenging air; $m_e$ is the mass flow of the exhaust gas; $dQ_w/dt$ is the rate of heat transfer; $dV_z/dt$ is the changing rate of the cylinder volume; $m_z$ is the gas mass in the cylinder; $U_z$ is the unit internal energy; $h_s$ is the unit enthalpy of scavenging air; $h_e$ is the unit enthalpy of exhausting gas; $p_z$ is the gas pressure in the cylinder; $V_z$ is the transient cylinder volume; $R_z$ is the gas constant in the cylinder; $T_z$ is the gas temperature in the cylinder; $c_{vz}$ is the specific heat of gas at constant volume; $LHV$ is the lower heat value of fuel oil.

For the calculation of the rate of heat transfer, the mass flow of scavenging air, and the mass flow of exhausting gas the method employed in [31] is followed. There is no further development in the present work. However, the thermodynamic properties of gas in the cylinder and the heat release rate of fuel developed further in the present work to satisfy the real time simulation.

### 2.1. Thermodynamic Properties of Gas in the Cylinder

The components of gas in the cylinder are complex, such as $CO_x$, $H_2O$, $NO_x$, $SO_x$, $O_2$, $N_2$, etc. For the sake of simplicity, gas in the cylinder is taken as compounds of pure air and pure product of combustion of oil. The components of the pure product of combustion depend on the fuel oil and the process of combustion. But, the molecule weight of the pure product is the only parameter we really care about, which can be obtained from experimental or theoretical calculation.

The mass fraction of burned fuel used to be taken as the parameter describing the composition of the gas. However, we never know how much fuel will be injected into the cylinder before the injection is finished. Hence, the mass fraction of burned fuel cannot be used here.

But we know how much pure product of combustion in the cylinder even during the fuel injection. Therefore, the fraction of pure product of combustion in the cylinder $\chi_g$ is used to describe the composition of gas in the cylinder, and it is defined as Equation (5).

$$\chi_g = \frac{M_g}{M_z}, \tag{5}$$

where, $M_g$ is the mass of pure product in the cylinder; $M_z$ is the mass of gas in the cylinder.

The generalized excess air ratio $\alpha_k$ can be defined as the ratio of the total gas mass over the mass of pure product, as shown in Equation (6).

$$\alpha_k = \frac{M_z - M_f}{-M_g} = \frac{1}{\chi_g} - \frac{1}{a_{stoich}}, \tag{6}$$

where, $m_f$ is the mass of the un-combusted fuel in the cylinder; $a_{stoich}$ is the stoichiometric air–fuel ratio, which depends on the fuel.

The gas constant and molecule weight of gas in the cylinder are derived by using the least square method based on the experimental data.

$$R_z = 287.0867 - 0.3944/\alpha_k, \tag{7}$$

$$M_z = 28.9705 - 0.0403/\alpha_k. \tag{8}$$

The specific heat ratio of ideal gas is a function of gas temperature, and the transient specific heat ratio of gas $\gamma_z$ in the cylinder can be calculated by Equation (9), which is derived from experimental data.

$$\gamma_z = 1.4373 - 1.318 \times 10^{-4} T_z + 3.12 \times 10^{-8} T_z^2 - 4.8 \times 10^{-2}/\alpha_k. \tag{9}$$

Therefore, we can get the specific heat at constant volume and the specific heat at constant pressure using the thermodynamic equation.

$\chi_g$ varies over time, because the mass of the pure product and the total gas varies during the combusting and gas exchanging processes.

$$\frac{d\chi_g}{dt} = \left( M_z \frac{dM_g}{dt} - M_g \frac{dM_z}{dt} \right)/M_z^2 = \frac{1}{M_z}\left( \frac{\alpha_{stoich}}{LHV}\frac{dQ_f}{dt} + \chi_{g,s} m_s - \chi_g m_e - \chi_g \frac{dM_z}{dt} \right), \tag{10}$$

where, $\chi_{g,s}$ is the fraction of the pure product of combustion in the scavenging air.

In fact, Equation (10) has a different form in different processes, for example, $dQ_f/dt$ will be zero when it isn't in the combustion process.

### 2.2. Gas Exchanging Process

The gas exchanging process plays an important role for setting the initial conditions for the combustion process. There are three consecutive processes in a two-stroke diesel engine: the exhaust process, the scavenging process, and the later exhaust process. However, the scavenging process is the only process that determines the components of gas in the cylinder at the start point of combustion. That means $\chi_g$ is determined by the scavenging process.

There are several types of scavenging models [32], including the three dimensional model [33], multi-zone model [34], and single zone model [34]. Considering the time of calculation, the single zone model is selected here. However, the scavenging quality of the "perfect mixing" model is lower than the real scavenging, yet the quality of the "thorough sweeping" model is better than the real scavenging. But the stratified scavenging model is time-consuming. Therefore, the so-called "concentrated exhausting gas" scavenging model is used in the present work. The concept of the model is that the fresh air entering into the cylinder immediately mixes with the gas in the cylinder, but the exhausting gas contains more combustion product than the mixed gas in cylinder.

Therefore, Equation (11) is obtained by modifying Equation (10) in the scavenging process.

$$\frac{d\chi_g}{dt} = \frac{1}{M_z}\left( \chi_{g,s}\frac{dM_s}{dt} - \xi\chi_g m_e - \chi_g \frac{dM_z}{dt} \right), \tag{11}$$

where, $\xi$ is a coefficient to adjust the scavenging quality. The value of $\xi$ is equal to or more than unity. If $\xi = 1$, the model becomes a "perfect mixing" model.

The value of $\xi \chi_g$ may be larger than the value of $\chi_g$ at the beginning of the scavenging process. It's impracticable, so a certain limitation should be exerted on Equation (11), as shown in Equation (12).

$$\frac{d\chi_g}{dt} = \begin{cases} \frac{1}{M_z}\left(\chi_{g,s}m_s - \xi\chi_g m_e - \chi_g \frac{dM_z}{dt}\right), & \xi\chi_g < \chi_{g0} \\ \frac{1}{M_z}\left(\chi_{g,s}m_s - \chi_{g0} m_e - \chi_g \frac{dM_z}{dt}\right), & \xi\chi_g \geq \chi_{g0} \end{cases}, \tag{12}$$

where, $\chi_{g0}$ is the ratio of product at the starting point of the scavenging process.

Following the same logic, the exhausting gas temperature is derived from the energy conservation equation, shown as Equation (13).

$$T_e = \begin{cases} \frac{1}{c_{ve}}(c_{vz}\xi T_z - \xi c_{vs}T_s + c_{vs}T_s), & T_e < T_{z0} \\ T_{z0}, & T_e \geq T_{z0} \end{cases}, \tag{13}$$

where, $T_e$ is the temperature of exhausting gas; $c_{ve}$ is the specific heat of exhausting gas at constant volume; $c_{vs}$ is the specific heat of scavenging air at constant volume; $T_{z0}$ is the gas temperature at the starting point of the scavenging process.

Figures 1 and 2 show the simulation values of $\chi_g$ and $T_z$ with different $\xi$ during the scavenging process. It can be concluded that the value of $\xi$ is employed to adjust scavenging quality. The larger the value of $\xi$, the better the scavenging quality. But the effect of $\xi$ gets less important when the value of $\xi$ gets larger. As we can see from Figures 1 and 2, there is no change of the value of $\chi_g$ at the very beginning of the scavenging process, and the gas temperature decreases a little. This phenomenon shows that the performance of the model has some characteristics of stratified scavenging.

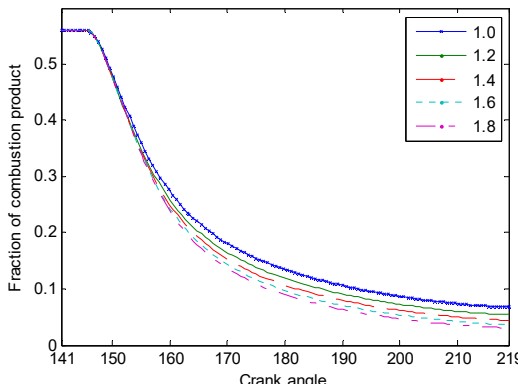

**Figure 1.** Simulation value of $\chi_g$ with different $\xi$ during the scavenging process. ($\chi_g$: composition of gas in the cylinder $\xi$: coefficient to adjust the scavenging quality).

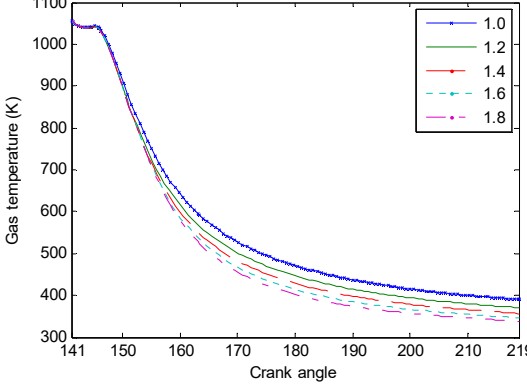

**Figure 2.** Simulation value of the gas temperature in the cylinder with different $\xi$ during the scavenging process.

## 3. Fuel Injection

Injection timing, profile, spray tip penetration, and spray cone angle of fuel injection are several important parameters of combustion. The fuel injection model should have the ability to calculate all these parameters.

### 3.1. Fuel Pressure in the Nozzle

Fuel pressure in the chamber of the nozzle is obtained by the continuity equation, as shown in Equation (14).

$$\frac{dp_n}{dt} = \frac{B_e}{V_n}\left(m_{F,n} - m_{F,inj}\right), \tag{14}$$

where, $V_n$ is the chamber volume of the nozzle; $B_e$ is elastic modules of fuel; $m_{F,n}$ is the fuel flow into the nozzle; $m_{F,inj}$ is fuel flow injected into the cylinder.

A common rail fuel injection system is taken as an example. There are injection valves to control the timing of the injection and the flow rate entering into the chamber of the nozzle. The fuel flow when passing through the fuel valve orifice is computed by Equation (15).

$$m_{F,n} = \mu_o A_o \sqrt{(p_r - p_n)\frac{2}{\rho_{F,r}}}, \tag{15}$$

where, $p_r$ is the oil pressure in the common rail; $A_o$ is the area of the fuel valve orifice; $\mu_o$ is the flow coefficient of the fuel valve; $\rho_{F,r}$ is the fuel density in the common rail.

### 3.2. Fuel Injection Rate

It's usual that there are several nozzle orifices in a nozzle, and the fuel flow injected into the cylinder should be the sum of the fuel flow of each nozzle orifice. Furthermore, there are several nozzles for one cylinder in some large scale engines, especially marine diesel engines, so the fuel flow should be the sum of all nozzles. However, the model for only one orifice is built in the present work for the convenience of description.

The mass rate of fuel injection is calculated by Equation (16).

$$m_{F,inj}(t) = \rho_{F,r} A_n u_{F,inj}, \tag{16}$$

where, $A_n$ is the cross section area of a nozzle orifice; $u_{F,inj}$ is the velocity of fuel oil spray, which is calculated by Equation (17).

$$u_{F,inj} = c_d \sqrt{\frac{2(p_n - p_z)}{\rho_{F,r}}}, \tag{17}$$

where, $p_z$ is gas pressure in the cylinder; $c_d$ is the discharge coefficient, which is calculated by applying the method shown in [35].

$$c_d = \frac{1}{\sqrt{c_{inlet} + f\frac{l}{d} + 1}}, \tag{18}$$

where, $l$ is the length of the orifice; $d$ is the diameter of the nozzle hole; $c_{inlet}$ is the inlet loss coefficient, which is a function of the ratio of the radius of the inlet round angle over the diameter of the nozzle hole, $r/d$, as shown in Figure 3; $f = \text{Max}(0.316Re^{-0.25}, 64/Re)$, $Re$ is the Reynolds number of fuel flow.

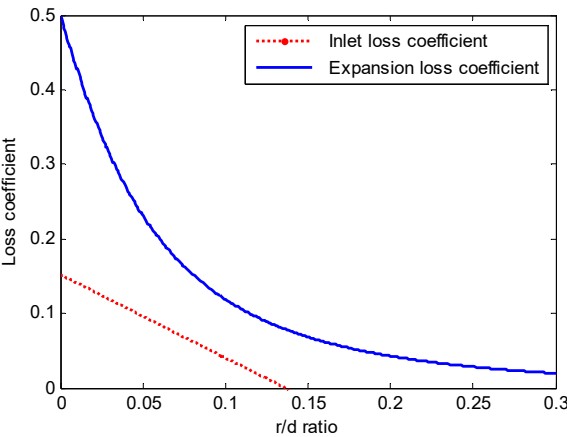

**Figure 3.** Inlet and expansion loss for different $r/d$.

### 3.3. Spray Model

Fuel spray has substantial effects on the mixing process, and thereby the combustion. There are two parameters of spray that affect the combustion process: spray cone angle and spray tip penetration.

The spray cone angle $\theta$ follows an aerodynamic-based model, which is expressed as

$$tan(\theta/2) = \frac{4\pi}{C} \sqrt{\frac{\rho_g}{\rho_{F,r}}} f(K), \tag{19}$$

where, $C$ is nozzle parameter, which is kept constant; $\rho_g$ is gas density in the cylinder; $f(K)$ is a function approximated by

$$f(K) = \sqrt{3}/6(1 - exp(-10K)), \tag{20}$$

where, $K$ is a function related to Reynolds number, Weber number, density of liquid fuel, and gas density.

However, the value of $f(K)$ can be seen as a constant when Equation (19) is applied to high pressure injections. The fuel injection velocity in calculating $K$ was kept constant at 120 m/s, but all the other parameters are still changing in literature [36]. Therefore, the value of $f(K)$ is obtained at an injection velocity of 120 m/s, and kept constant to reduce the complexity of the model.

The gas density in the cylinder is determined by the volume of the cylinder and the mass of the gas in the cylinder.

$$\rho_g = \frac{M_z}{V_z}. \tag{21}$$

The mass of the gas is mainly determined by the gas exchanging process, and slightly determined by the fuel injected into the cylinder. Therefore, the scavenging process affects combustion in many ways.

The spray tip penetration is derived as:

$$s(t) = \left[ \int \frac{4M(t)}{\pi/3k^* \rho_g tan^2(\theta/2)} \right]^{0.25}, \tag{22}$$

where, $k^*$ is a model constant, which is optimized to a certain nozzle and set at 0.6514 here; $M(t)$ is the integration of momentum flow of the fuel injection calculated by Equation (23).

$$M(t) = \int \dot{M}(t)dt = \int \rho_{F,r} A_n u_{F,inj}^2(t)dt. \tag{23}$$

## 4. Combustion Model

As soon as the fuel is injected into the cylinder, the evaporation of fuel starts. After a certain concentration of the fuel vapor is reached, a self-ignition will occur under the high temperature produced

by the compression process. Arrègle [36] divided the combustion process into four consecutive phases: pre-mixed combustion, initial transient diffusion combustion, quasi-steady diffusion combustion, and final combustion, and analyzed the effects of fuel concentration and oxygen concentration on apparent combustion time. Chmela [26] adopted a three combustion concept, as is shown in Figure 4. The fuel quantity injected into the cylinder before the ignition is called pre-mixed fuel, the combustion of which is called pre-mixed combustion. As soon as the self-ignition occurs, the two types of combustion start at the same point. Furthermore, the so-called diffusion combustion process is divided into two stages, not three. These stages are the burning spray stage and the burning-out stage. The former refers to the combustion process during the injection of fuel, yet the later refers to the combustion process after the injection is finished.

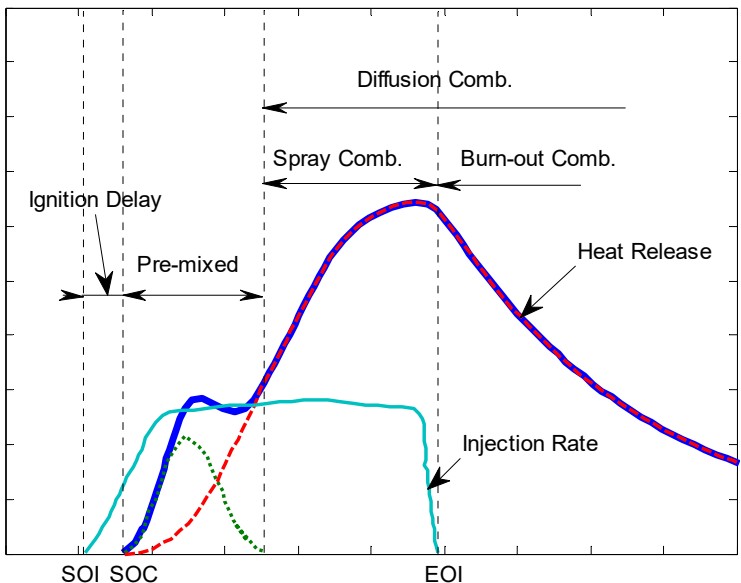

**Figure 4.** Phases of combustion in a diesel engine.

The heat release rates in different combustion phases are determined by the different reaction rates. The reaction rates are usually determined by different types of equations in different combustion phases. Therefore, the total heat release rate is the sum of the heat release rates in different combustion processes.

However, the model may encounter some troubles when post-injection or multi-injections occur. At the time of post-injection occurring, it must be in the burn-out combustion stage. Will the fuel injected into the cylinder by post-injection burn according to the spray combustion model, or the burn-out combustion model? Each stage is probably good, but the model will inevitably become more complicated because the model has to distinguish post injections from the main injection, yet they have no difference in physics.

We can see the evolution of fuel spray during combustion in literature [37]. The spray maintains its shape after the injection is finished. Therefore, it is reasonable that the same reaction model is used in the whole diffusion combustion phase. We do not differentiate the two stages of diffusion combustion. So, a total heat release rate equation can be obtained as Equation (24).

$$\frac{dQ_{tot}}{dt} = \frac{dQ_{pre}}{dt} + \frac{dQ_{diff}}{dt},$$ (24)

where, $Q_{tot}$ is the total released heat of fuel; $Q_{pre}$ is the released heat of the pre-mixed combustion; $Q_{diff}$ is the released heat of the diffusion combustion.

### 4.1. Reaction Rate Model

The reaction rate model should reveal the influence of the environmental features and the characteristics of fuel in a simple way. The gas turbulence and the gas temperature in the cylinder are the environmental features selected to describe the reaction rate. The activation energy of fuel is the characteristic of fuel selected to control the reaction rate. A Magnussen type of equation, Equation (25), and Arrhenius type of equation, Equation (26), are employed here to describe the turbulence controlled rate and the temperature controlled rate respectively.

The Magnussen type reaction rate $r_{Mag}$ is calculated by

$$r_{Mag} = c_{Mag} c_R \frac{\sqrt{k}}{\sqrt[3]{V_z}}, \tag{25}$$

where, $c_{Mag}$ is a model constant; $c_R$ is the concentration of the rate controlling substance; $k$ is the turbulent kinetic energy density.

The Arrhenius type reaction rate $r_{Arr}$ is given by

$$r_{Arr} = c_{Arr} c_F c_o e^{\frac{-E_a}{R_z T_z}}, \tag{26}$$

where, $c_{Arr}$ is a model constant; $c_F$ is the concentration of fuel in the spray; $c_o$ is the concentration of oxygen in the spray; $E_a$ is the activation energy of fuel, which shows the influence of the fuel quality on the reaction rate.

The gas temperature in the cylinder $T_z$ is used here to calculate the reaction rate. However the temperature in the burning spray is much higher than that in the cylinder. Therefore, there must be a coefficient $c_T$ to compensate for the difference, which is the ratio of the mass in the cylinder over the mass in the spray. Hence, $T_z$ in Equation (26) is replaced by $c_T T_z$.

Somehow, the reaction rate $r_{Mag}$ can be seen as an evaporation rate of fuel, yet the reaction rate $r_{Arr}$ can be seen as the chemical reaction rate of fuel. The two mechanisms occur consecutively. Therefore, the total characteristic time of the reaction is the sum of the two characteristic times consumed by the two mechanisms respectively. Furthermore, we can get the equation of the total reaction rate $r_{tot}$ as Equation (27).

$$r_{tot} = \frac{1}{\tau_{tot}} = \frac{1}{\frac{1}{r_{Mag}} + \frac{1}{r_{Arr}}} = \frac{r_{Mag} r_{Arr}}{r_{Mag} + r_{Arr}}. \tag{27}$$

#### 4.1.1. Concentrations of Fuel and Oxygen

The volume of the spray is crucial to the calculation of the reaction rate. Based on literature, the spray is somehow like a cone. Therefore, the volume of the spray can be approximated by Equation (28).

$$V_{mix} = \frac{\pi}{3}\left(s^3 tan^2\frac{\theta}{2} + \frac{3}{2}d_n s^2 tan\frac{\theta}{2} + \frac{3}{4}d_n^2 s\right). \tag{28}$$

But the mass of gas entrained into the spray is required to be calculated. It was obtained by the air–fuel excess ratio in the mixture cloud, which is assumed to have a certain spatial distribution. Then, the mean concentration of fuel and oxygen is obtained by integration in the mixture cloud [26]. However, the ratio of the pure combustion product in the cloud isn't considered. Furthermore, the model requires the density of fuel vapor, which is difficult to obtain.

To accelerate the speed of computing, the spray is taken as a whole. A coefficient is introduced to indicate the volume ratio of the entrained gas to the whole spray. The mean air–fuel excess ratio $\lambda$ in the spray is obtained with the consideration of the ratio of the combustion product in the spray.

$$\lambda = \frac{\rho_g(1 - \chi_g)(V_{mix} - M_F/\rho_F)}{M_F \alpha_{stoich}}, \tag{29}$$

where, $c_{g,en}$ is the coefficient of the entrained gas; $M_F$ is the fuel mass contained in the spray, which is calculated by

$$M_F = \int m_{F,inj} dt. \tag{30}$$

A relationship is built between the quality of scavenging process and the combustion process by Equation (29).

Therefore, the mean concentration of fuel and oxygen are calculated by Equation (31) and Equation (32) respectively.

$$c_F = \frac{M_F}{V_{mix}}, \tag{31}$$

$$c_o = \frac{\lambda M_F}{V_{mix}}. \tag{32}$$

Let's review $c_R$ in Equation (25). It is either the concentration of oxygen or that of fuel depending on whether the air–fuel excess ratio is higher or lower than unity:

$$c_R = \begin{cases} c_F, & \lambda \geq 1 \\ c_o, & \lambda < 1 \end{cases}. \tag{33}$$

It can be found from Figure 5 that there is a comparatively small difference between the two curves at the very beginning of the injection and combustion processes, but the difference gets larger as more air is entrained into the spray and more fuel is combusted. The initial difference is due to the value of $\chi_g$ at the end of the scavenging process. The reason why the difference is changing is that the value of $\chi_g$ is changing during combustion. The difference in the concentration of oxygen between the two curves is changing following the same law, as shown in Figure 6. This shows that the value of $\chi_g$ can affect the reaction rate, and in turn the combustion process. Therefore, the quality of scavenging will influence the quality of the combustion process in the model, as shown in Figures 7 and 8 respectively. The larger the value of $\xi$, the larger the values of $\lambda$ and $c_o$, and the better the quality of combustion will be.

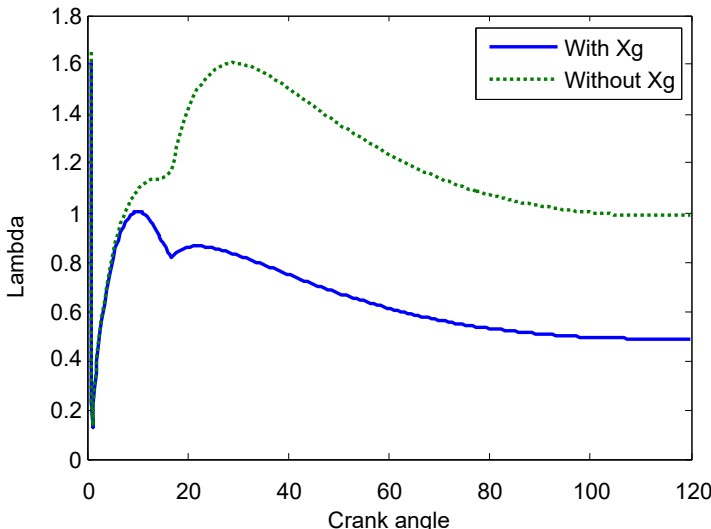

**Figure 5.** Curves of the value of $\lambda$ with and without $\chi_g$ during combustion.

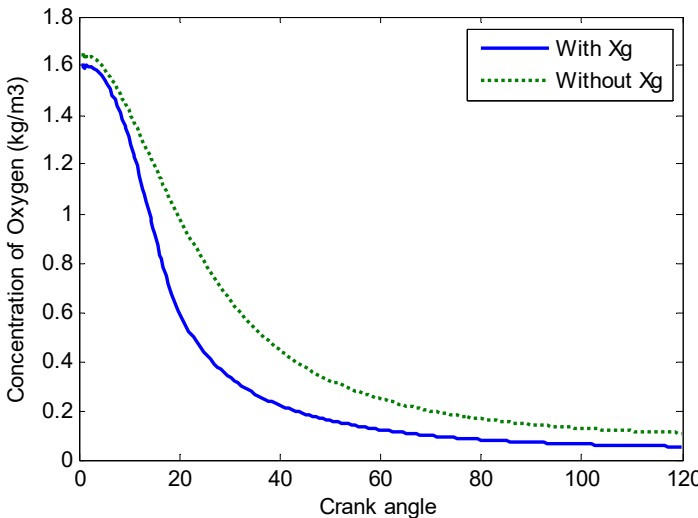

**Figure 6.** Curves of the value of $c_o$ in the fuel mixture with and without $\chi_g$ during combustion.

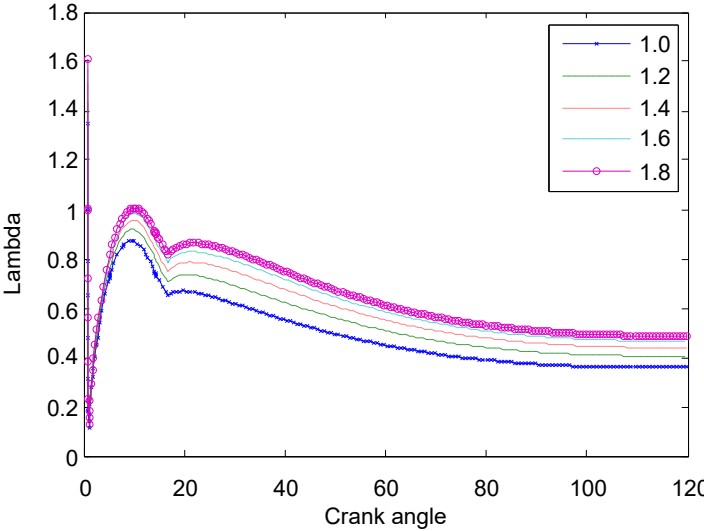

**Figure 7.** Curves of the value of $\lambda$ with different values of $\xi$.

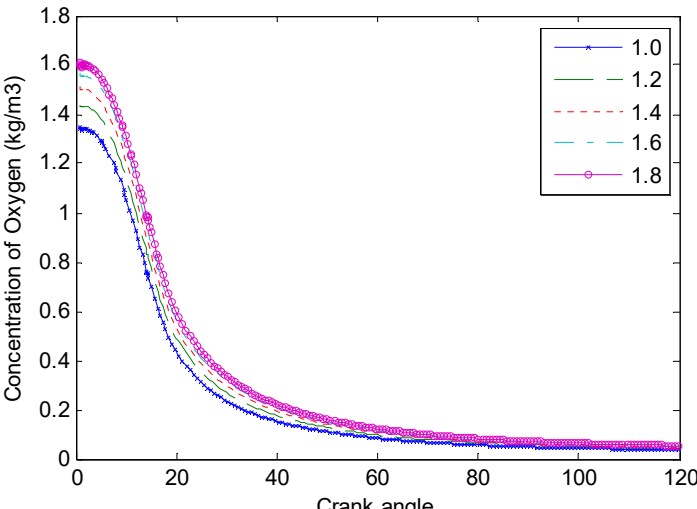

**Figure 8.** Curves of the value of $c_o$ with different values of $\xi$.

### 4.1.2. Kinetic Energy

There are three sources of turbulent kinetic energy: the kinetic energy of the intake flow, the squish flow out of the space between the piston and the cylinder head into the combustion bowl, and the kinetic energy of the injection sprays. But the kinetic energy of the injection sprays is the only important source of turbulence [26].

The amount of the kinetic energy $E_{kin,F}$ charged into the cylinder by the injection spray is determined by injection mass flow and its velocity, as shown in Equation (34).

$$\frac{dE_{kin,F}}{dt} = 0.5 m_{F,inj} u_{F,inj}^2. \tag{34}$$

The kinetic energy will dissipate in a certain period of time. A coefficient $c_{Diss}$ is used to adjust the dissipation rate. Therefore the transient kinetic energy is calculated by Equation (35).

$$\frac{dE_{kin,F,Diss}}{dt} = \frac{dE_{kin,F}}{dt} - c_{Diss} E_{kin,F,Diss}. \tag{35}$$

Then the kinetic energy density is approximated by Equation (36).

$$k = c_{Turb} \frac{E_{kin,F,Diss}}{m_z}, \tag{36}$$

where, $c_{Turb}$ is considered as a transfer efficiency. The coefficient $c_{Turb}$ is less than unity, which means that only a part of the kinetic energy is transferred to the substance in the cylinder. The instantaneous kinetic energy density approximated by Equation (36) is a little less than the real value, because the total mass of the gas in the cylinder is taken as the denominator.

### 4.2. Ignition Delay

The ignition delay model is used to determine the time between the start of the injection (*SOI*) and the start of the combustion (*SOC*). The time delay depends on the preparation process of the combustible gas mixture, such as fuel vaporization, fuel-air mixing, and chemical pre-reaction. The total reaction rate is employed here to determine the preparation process. When the integration of $r_{tot}$ hits the a threshold value *I*, the combustion starts.

$$\int_{SOI}^{SOC} r_{tot} dt = I, \tag{37}$$

where, *SOI* is the starting time of the injection; *SOC* is the starting time of the combustion; *I* is a threshold value to adjust the ignition delay time. The value of *I* can also adjust the amount of fuel that will be used in pre-mixed combustion, and in turn affects the maximum pressure in the cylinder.

### 4.3. Pre-Mixed Combustion

Pre-mixed combustion refers to the combustion of fuel mixtures prepared during the ignition delay. The total reaction rate is taken as the reaction rate in pre-mixed combustion. Therefore, an equation for pre-mixed combustion is obtained as:

$$\frac{dQ_{pre}}{dt} = r_{tot} M_{F,pre} LHV (t - t_{SOC})^2, \tag{38}$$

where, *LHV* is the low heat value; $M_{F,pre}$ is the available fuel mass for pre-mixed combustion; *t* is the elapsed time from the starting time of injection; $t_{SOC}$ is the starting time of combustion; $(t - t_{SOC})$ is the elapsed time from the starting of the combustion.

Along with the pre-mixed combustion, the available fuel mass is reducing, as shown by Equation (39).

$$M_{F,pre}(t) = \int_{SOI}^{SOC} m_{F,inj}dt - \frac{Q_{pre}(t)}{LHV}. \tag{39}$$

### 4.4. Diffusion Combustion

At the start of combustion, the diffusion combustion starts. That means that there are two types of combustion at the same time, and the available fuel mass is divided into two parts. One part is the $M_{F,pre}$ shown in Section 4.3. The other is the available fuel mass in the diffusion combustion process calculated by Equation (40).

$$M_{F,diff,avail} = \int_{SOC}^{t} m_{F,inj}dt - \frac{Q_{diff}}{LHV}. \tag{40}$$

At the end of injection, only around 50% of the fuel injected into the cylinder is burnt. So, the combustion continues after the injection is finished. The same equation is used to calculate the heat release rate in the two stages, shown as Equation (41).

$$\frac{dQ_{diff}}{dt} = c_{Mod}r_{tot}LHVM_{F,diff,avail}, \tag{41}$$

where, $c_{Mod}$ is a model constant; $M_{diff,avail}$ is the fuel mass that can be used in the diffusion combustion.

The total reaction rate $r_{tot}$ is applied in Equation (41). But if the temperature is high enough, the reaction rate derived from the Arrhenius type equation will be high enough. Then, the total reaction rate $r_{tot}$ is determined by the Magnussen type of equation. That can be concluded from Equation (27).

If multi injections occur, the fuel quantity and the kinetic energy injected into the cylinder during multi injections will be calculated by the same equations. The model needs no extra mechanism added to be adapted for multi injections. Figure 9 shows schematically the heat release rate profile when multi injections occur during the whole combustion process. The reasonable sharp changes can be seen when a new stage of injection begins in Figure 9.

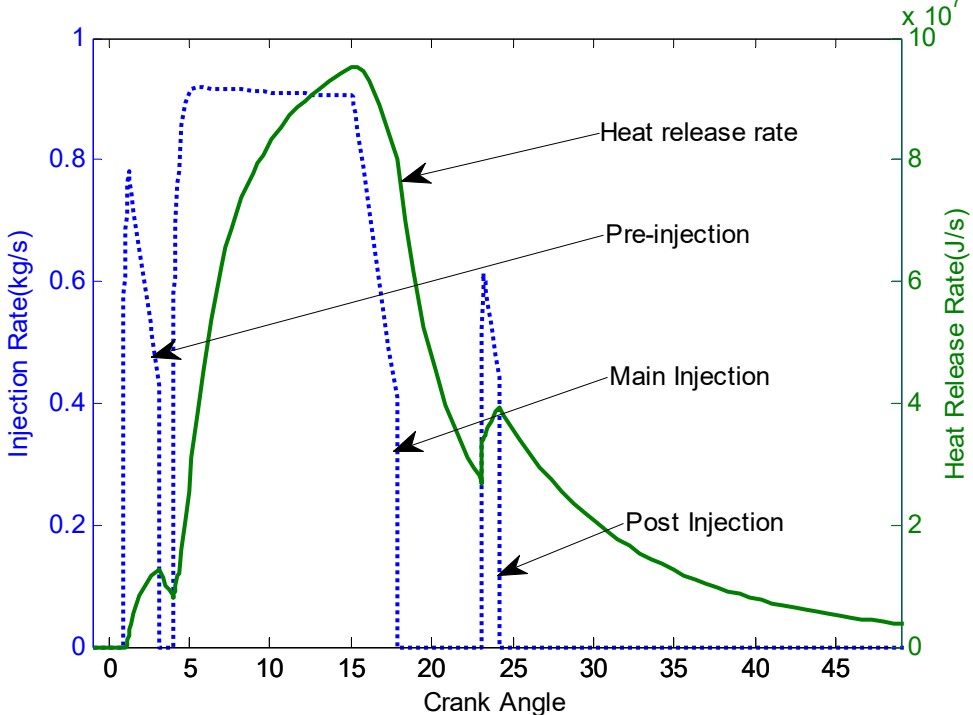

**Figure 9.** Heat release rate when a post injection occurs.

## 5. Results and Discussion

### 5.1. Test Engine

A large-scale two-stroke marine diesel engine is taken as the test engine. The fuel system of the test engine is a high pressure common rail system, in which the injection is controlled by a set of rail valves. The ECU (engine control unit) controls the timing of the rail valves and the quantity of fuel injected into the cylinder. Three nozzles are allotted for each cylinder, and one, two, or three nozzles may work depending on the engine load. The fuel rail pressure varies with different loads and speed of the engine. The scavenging process of the test engine is the uniflow type, and the timing of the opening and closing of the exhausting valve is adjusted depending on the load and the speed of the engine. The main data of the engine is shown in Table 1.

**Table 1.** Data of the test engine.

| Item | Value |
|---|---|
| Number of Cylinders | 7 |
| Stroke (m) | 2.292 |
| Bore (m) | 0.6 |
| Rated Power (kW) | 11,650 |
| Rated Speed (r/min) | 114 |
| Compression Ratio | 18.4 |

For the simulation model, the concentration exhaust coefficient in the model is set to 1.4, the kinetic energy dissipation rate parameter is 0.1, the transmission efficiency coefficient is 0.8, the combustion rate constant is 0.8, the threshold of combustion starting point is 0.5, and other parameters are selected according to the actual equipment. When the simulation conditions change, the values of each parameter remain unchanged. In fact, the different concentration exhaust coefficient values will affect the simulation results. For example, for the calculation results of exhaust temperature, through the comparison of simulation and test results, the concentration exhaust coefficient is finally set to 1.4, but for low-speed marine diesel engines with different working parameters, the empirical parameters may be different.

### 5.2. Simulation Results

Two cases of the engine running conditions are simulated. Parameters of the two cases are shown in the Table 2. The traditional single zone first law heat release model is employed to calculate the heat release rate from the experimental data.

**Table 2.** Parameters of the two cases of the engine running conditions.

| Item | Case 1 | Case 2 |
|---|---|---|
| Temperature of scavenging air (°C) | 30 | 36.2 |
| Engine speed (r/min) | 90.5 | 114 |
| Engine load (%) | 50 | 100 |
| Maximum pressure (bar) | 106.7 | 153.9 |
| Compression pressure (bar) | 77 | 136.6 |
| Pressure of scavenging air (bar) | 1.93 | 3.65 |
| Pressure in exhaust manifold (bar) | 1.65 | 3.11 |
| Fuel pressure in common rail (bar) | 600 | 900 |
| Timing of injection (CA) | −2.3 | 0.8 |

Figures 10 and 11 show the indicator diagrams for the two cases respectively. The simulation results agree with the experimental data well, although there are some slight deviations during the

combustion process. In fact, the consistency between the simulation results and the experimental data is good enough for engine performance prediction, especially for the simulation of the control system.

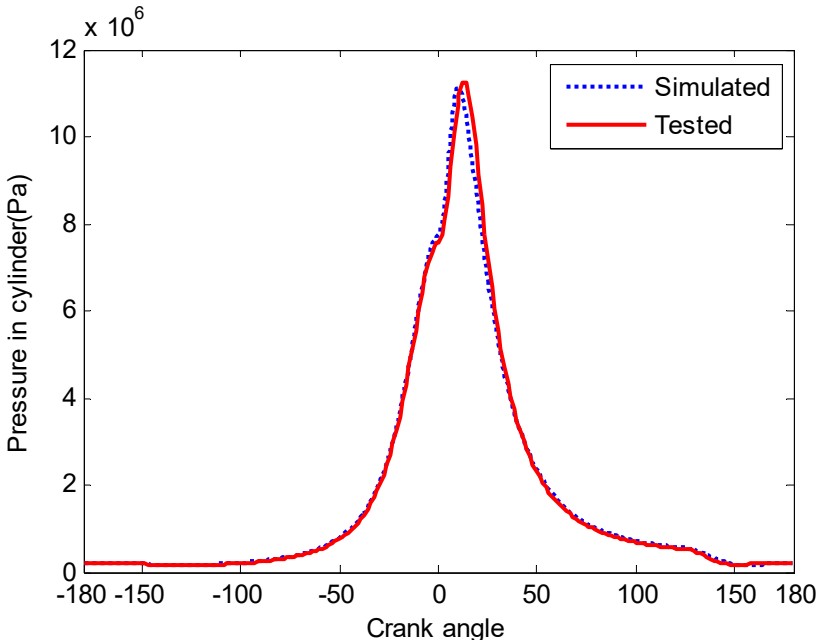

**Figure 10.** Indicator diagram of case 1.

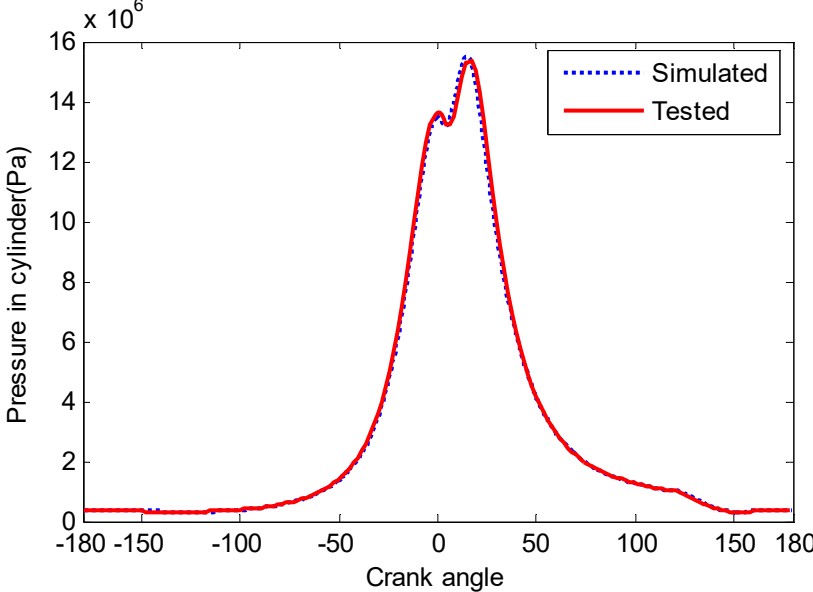

**Figure 11.** Indicator diagram of case 2.

Figures 12 and 13 show the heat release rate and the injection rate for case 1 and case 2 respectively. Although the analyzed curve isn't as smooth as the simulated curve, they agree in general. Since the traditional single zone first law heat release model is employed to analyze the heat release rate, more heat transfer through the wall of the combustion chamber is added to the heat release during analysis, because there is no further distinguishing of the heat transfer owing to combustion from the total heat transfer. Furthermore, the calculated pressure differential during analysis is larger than the real one. Therefore, the shape of the analyzed curve is "fatter" than the simulated curve, and the simulated ignition delay is slightly larger than the analyzed one. But the peak of the simulated heat release rate

is the same as that of the analyzed. Therefore, the performances of the engine in different running conditions can be accurately simulated by the model.

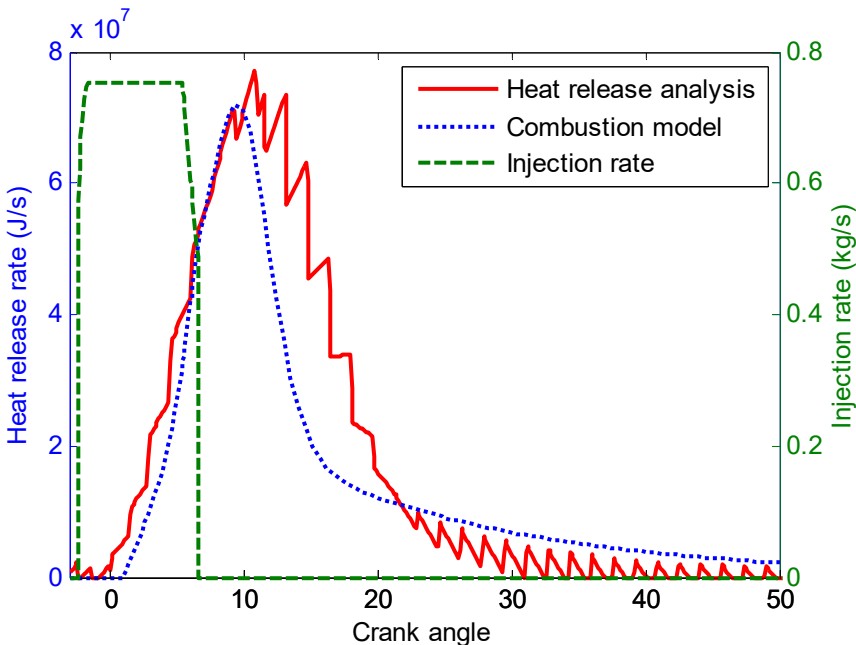

**Figure 12.** Heat release rate of case 1.

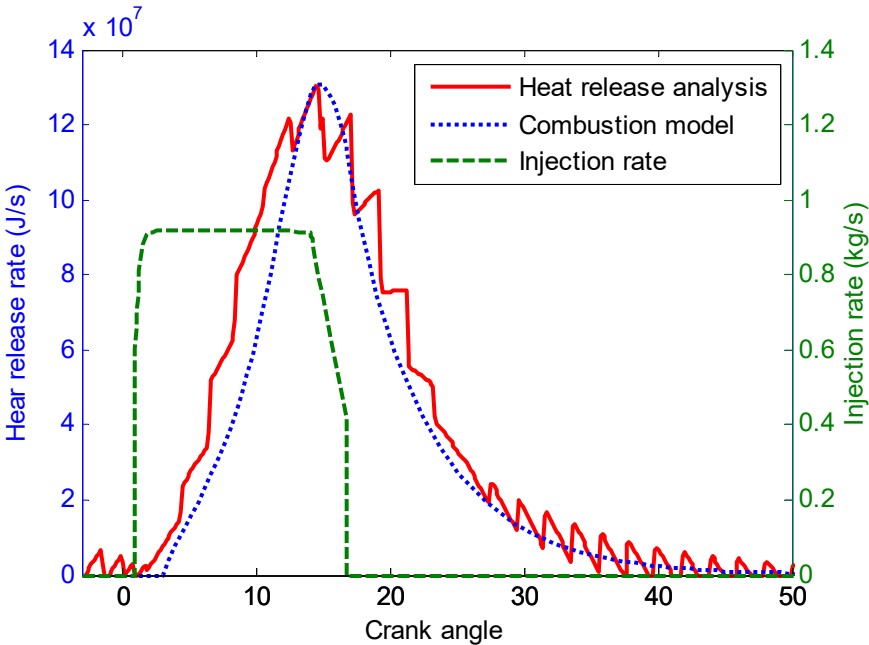

**Figure 13.** Heat release rate in case 2.

### 5.3. Speed of Computing

Simulations are performed on a personal computer, which has an Intel®Core™i3-7100 3.9G CPU and a 4GB DDR memory. The operating system is Windows 7. The engine model is built in Matlab/Simulink (R2016b, Shanghai, China), as shown in Figure 14. The name of each module shows its function. The "Loop" module only acts as a passage for signals, in which the signal will be further handled if it may form an "Algebraic Loop". The "CombProc" module is employed here to provide the enable signal for the "Combustion" module. The "ECU" module acts as controller and its acting

mechanism. FOCR stands for fuel oil common rail. VCU stands for valves control unit, which controls the exhaust valves. ICU stands for injection control unit, which controls the fuel injection process. These three modules are less significant in this paper. Others are easy to understand by their names, so we don't explain them further.

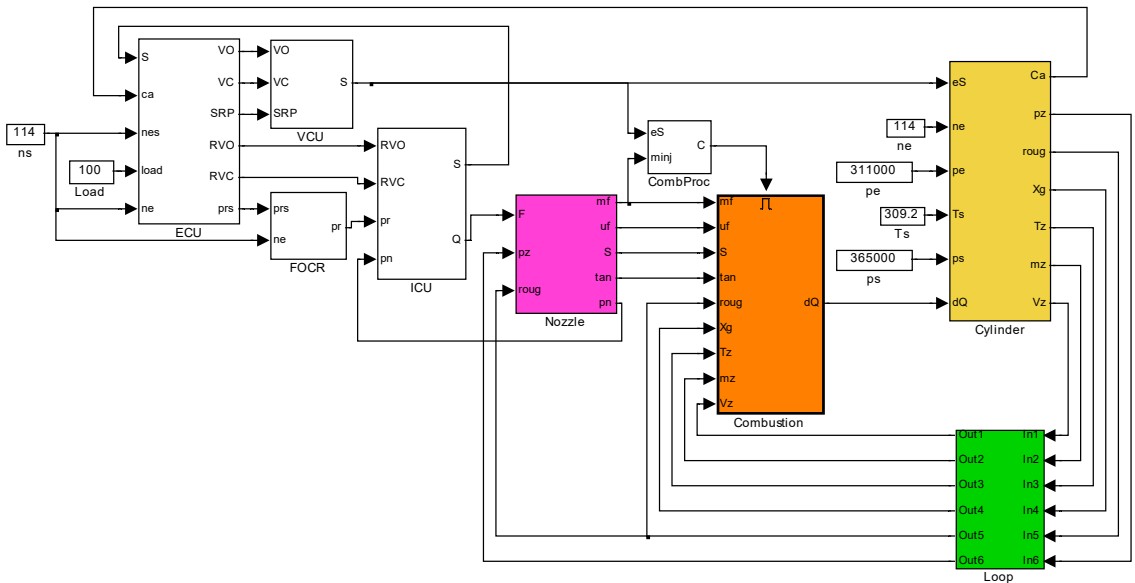

**Figure 14.** Simulink model for the combustion model.

The physical process of the model was set to 60 s. The model was run 5 times for each case, as shown in Table 2, and the results are shown in Table 3. The mean value of the elapsed times for case 2 is 15.04 s. Therefore, the model has the potential to be applied in real time engine performance simulation for low speed and medium speed diesel engines.

In fact, models for the intake manifold, exhaust manifold, turbocharger, and crank motion are several extra modules required by a dynamic engine model. These modules wouldn't much increase the load on computing, because the time consumed is largely due to the small time step size required in the combustion process.

**Table 3.** Elapsed time for simulating the model.

| No. | 1 | 2 | 3 | 4 | 5 | Mean |
|-----|-----|-----|-----|-----|-----|------|
| Elapsed time (s) (Case 1) | 14.41 | 14.59 | 14.75 | 14.87 | 15.07 | 14.74 |
| Elapsed time (s) (Case 2) | 14.82 | 14.79 | 14.79 | 15.57 | 15.22 | 15.04 |

## 6. Conclusions

A simple scavenging model "concentrated exhausting gas" is modified and employed. The larger the concentration coefficient of the model, the better the quality of scavenging process. A transmission mechanism from the quality of the scavenging process to the quality of combustion is also established by employing the fraction of the combustion products to calculate oxygen concentration in the combustion model. The better the quality of the scavenging process, the faster the reaction rate of the combustion. Furthermore, the larger the coefficient of the scavenging model, the better the process of combustion.

A zero-dimensional combustion model is modified and applied in calculating the heat release rate of diesel engines, which combines the two stages in the diffusion combustion phase into one. The combination doesn't deteriorate the performance of the combustion model, yet provide a simple method to simulate the combustion process with multi injections of fuel. Due to the increase in

parameters that can be adjusted in contrast to the Wiebe's function, the model can satisfy more requirements of different simulation applications, and in turn be used in more situations.

The combustion model can work well with the model of the working process in the cylinder. Simulation data, including an indicator diagram and the heat release rate of combustion, agree with the experimental data very well in different running conditions of the engine. Moreover, the speed of computing is fast enough to be applied in real time applications for low and medium speed engines. Hence, the fast engine model presented in this paper can be used in real time simulations, such as control system simulation, performance simulation, and so on.

**Author Contributions:** Y.F. and H.W. contributed in equal parts to the establishment of the model, the calculation and the writing, whereby R.G. and Y.Z. had some useful suggestions and been involved in the discussion and preparation of the manuscript.

**Funding:** We gratefully acknowledge the financial support of the National Key Research and Development Program of China (No. 2016YFC0205400) and the Provincial Funding for National Projects of Heilongjiang Province in China (No. GX17A020).

**Conflicts of Interest:** The authors declare no competing financial interest.

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
