# Peer review of "A Zero-Dimensional Mixing Controlled Combustion Model for Real Time Performance Simulation of Marine Two-Stroke Diesel Engines"

_energies, doi:10.3390/en12102000_

Round 1
Reviewer 1 Report
Aim of the paper is to present the performance of a prediction model of marine low-speed two-stroke diesel engine operation based on an advanced mixture controlled combustion model, named “concentrated exhausting gas” model. The full model is presented extensively and fully, in its innovative aspects as well as the more consolidated parts. Its effectiveness is tested against real data. However, a comparison between the performance of the new model against the “classic” ones and related discussion is not performed, and this represents the most important lack of the manuscript. In order the paper to be eligible for publication, the Authors shall focus on these aspects. Minor points are:
Line 30: please replace “can achieve” with “can be achieved”
Page 3: the number associated to formulas are wrong (1 is repeated twice); the first law of thermodynamics (first formula 1) shall be well motivated: is it written in “open” or “closed” system conditions?
Line 109: the word “ingredient” in this contest sounds unappropriate;
Figure 5: please replace “lamda” with “lambda”;
Lines 422-425: please rephrase the sentence
Author Response
Dear reviewer, Thank you very much for the excellent and professional revision of our manuscript. The manuscript is entitled as follows: “A zero-dimensional mixing controlled combustion model for real time performance simulation of marine two-stroke diesel engine”. We have carefully revised the manuscript according to your suggestion. Our responses to the comments are listed below: Comment: Aim of the paper is to present the performance of a prediction model of marine low-speed two-stroke diesel engine operation based on an advanced mixture controlled combustion model, named “concentrated exhausting gas” model. The full model is presented extensively and fully, in its innovative aspects as well as the more consolidated parts. Its effectiveness is tested against real data. However, a comparison between the performance of the new model against the “classic” ones and related discussion is not performed, and this represents the most important lack of the manuscript. Reply: Thank you very much for your suggestion of amendment, which has been revised and perfected accordingly. As the reviewer pointed out, the overall language expression of the paper needs to be further refined. It may be that the expression in my paper is not clear. In fact, the mixture controlled combustion (MCC) model is used to simulate combustion characteristics in cylinder, while the centralized exhausting gas model is used to simulate residual exhaust characteristics in engine scavenging process. The two models are two different models. Combining the two models, the whole working process of the engine can be simulated. Therefore, the relevant calculation contents of the classical model and the concentrated exhausting gas model proposed by the reviewer can be referred to figs. 1 and 2 (If_=1, the model becomes a "perfect mixing" model.). In addition, for low-speed marine diesel engines with different excess air coefficients, the scavenging perfection will be different, and the parameters of the scavenging model will be different, so in the end, the author additionally lists the values of some model parameters for the simulation model. Based on that, we have revised our manuscript. The main modifications are added in the 413th~421th lines, and are as follows: For the simulation model, the simulation model, the concentration exhaust coefficient in the model is set to 1.4, the kinetic energy dissipation rate parameter is 0.1, the transmission efficiency coefficient is 0.8, the combustion rate constant is 0.8, the threshold of combustion starting point is 0.5, and other parameters are selected according to the actual equipment. When the simulation conditions change, the values of each parameter remain unchanged. In fact, the different concentration exhaust coefficient values will affect the simulation results. For example, the calculation results of exhaust temperature, through the comparison of simulation and test results, the concentration exhaust coefficient is finally set to 1.4, but for low-speed marine diesel engines with different working parameters, the empirical parameters may be different. Additionally, we have checked English grammar, spelling and sentence structure carefully in the revised manuscript. Thank you and best regards. Yours sincerely, Yongming Feng

Reviewer 2 Report
The paper is well written and structured in terms of scientific aspects and also brings novel and interesting topic in the field of IC Engine. I believe no further comments are needed to be considered in the paper and it can be published in the current stance. Thank you.
Author Response
Dear reviewer, Thank you very much for the excellent and professional revision of our manuscript. The manuscript is entitled as follows: “A zero-dimensional mixing controlled combustion model for real time performance simulation of marine two-stroke diesel engine”. We have checked English grammar, spelling and sentence structure carefully in the revised manuscript. Thank you and best regards. Yours sincerely, Yongming Feng

Reviewer 3 Report
(Attached you have a PDF with these comments, to view them better).
The authors present an interesting modelling approach for real time combustion prediction in a 2S marine diesel engine. The research is relevant, and the topic is interesting and valuable. However, there are some concerns regarding the model assumptions, as well as on the English style of the manuscript, as I will try to justify with my comments.
Here you have some major comments:
· The English style and grammar of the whole manuscript need to be revised. I will put some examples of problems in some sentences in the list of minor comments, at the end of this list.
· Line 92. The second equation in the manuscript is identified by (1), as the first equation. Consequently, the identification of all the equations is wrong. Authors should revise these identifications of the equations, as well as all the references to the equations appearing in the text.
· Line 124. I don’t agree with this equation of the generalized excess air ratio, . If the equation is developed, you find that , where is the mass of pure product in the cylinder (i.e. stoichiometric burned gases), and is the un-combusted fuel. This is incorrect! What is correct is , but this is not what your equation is stating!!
· Equations in lines 129 and 130. There are several concerns about these equations. On the one hand, the R of a particular gas (Rz in this case) is obtained by dividing the universal R (Â) by the molecular weight of the gas: . The two equations given by the authors are not consistent with this relationship!! If we represent the evolution of Rz based on the authors’ equation (Rz authors) and the one coming from the equation I’ve just introduced (which depends on Mz), in both cases versus the generalized excess air ratio (), the trends are not the same (see figure below):
· On the other hand, I think it is not a good idea to use the constant g (acceleration of gravity) in the equation for Rz, since it makes no physical sense. If g has some relationship with Rz, it should be the reverse relationship: Rz is not proportional to g, but conversely proportional to g (or proportional to 1/g), because the mass is in the denominator of the equation I’ve indicated above. So, the equation given by the authors is misleading…
· Line 177. In this graph, the authors show the impact of x on the combustion evolution. But unfortunately they don’t indicate which the value of the parameter that they finally use in their calculations is. I will latter comment a little bit more about this.
· Line 186. I assume that “continuous equation” should be “continuity equation”.
· Line 187. I’m a little bit confused with this equation. If Vn is the volume of the chamber volume of the nozzle, this is a constant magnitude. Why, then, dVn/dt appears in the equation?? This derivative should be 0!!
· Line 203. In this equation, there should be a density multiplying in the term of the right of the equation. Otherwise, the units are not consistent in the equation.
· Line 220. In the equation proposed by the authors, the ambient density () affects the spray cone angle with an exponent 0.5 (a square root). This value seems to be too high. If you look at reference SAE 960034, from Naber and Siebers, they talk about an exponent of 0.19. Why have you used such a big exponent??
· Line 269. I’m confused about this “reaction rate model”. If you are developing a combustion model which is mixing controlled, what’s the purpose of this “reaction rate model”?? Usually, in a MCC what you assume is that combustion takes place once mixing has taken place. This is much simpler than what you are doing (and more realistic). Any comment on that??
· Line 318. Still one more confusing thing… If you have a mixing controlled combustion, which is a diffusive combustion, under these circumstances the combustion always takes place under near-stoichiometric conditions. Why, then, the authors introduce a discussion depending of the lambda of the combustion process?? This discussion is not relevant here, because combustion always takes place at lambda » 1!!
· Figures 12 and 13. Looking at the experimental HRR, you should probably filter a little bit more the signal, to avoid this “ugly” oscillations in the curves.
· Line 440. The authors state that no further explanation about the Simulink model presented in figure 14 is needed. But I have still some doubts of the role of some of the boxes: what is FOCR, VCU and ICU??
· When showing the results for 2 different cases (i.e. two operating conditions), nothing is said about the set of model constants (for instance, x). Are they the same for both cases?? Or are they different?? This is very important to be known, since this would give more information about the prediction capability of the model.
· Line 443. Finally, you present only results for Case 2, regarding the calculation speed. As far as I understand, you are simulating something that takes place in 60 seconds in 35.5 s (in average), isn’t it? This is not very clear in the text. If you say that “the simulation time was set to 60 seconds”, it is not very clear if this is the time for the physical process, or if this is the time for the simulation. Please, clarify. And the other thing is that it would be nice if you show the same information but for Case 1 (even if it is probably less exigent –because of the lower engine speed-), since this will “enrich” the information given in the paper regarding suitability of the model for real time calculations.
And here you have some minor comments:
· Abstract, line 19. “…the speed of calculation is faster enough…” In this sentence, “faster” should be “fast”.
· Line 33: “…two types of model”. “model” should be “models”, since you are referring to more than 1 model type…
· Line 37-38: “…and other parameters related to combustion process” should be “…and other parameters related to the combustion process.” (a “the” should be added).
· Line 47-48. “Therefore, we have to resort to thermodynamic model.” This sentence is incorrect. I cannot get the meaning…
· Line 65. “…are applied in many a case” should be “…are applied in many cases”.
· Line 82-83. “The present work is initially to find a solution for these two problems”. This sentence is incorrect (incomplete). Perhaps you are referring to “The objective of the present work…”
· Line 97 (and in many other places). “…heat releasing rate…” should be “…heat release rate…”
· Line 105. “…is followed the method employed in [31]” should be “…the method employed in [31] is followed”.
· Line 137-138: “… the total gas varies during they are combustion process…” is incorrect. I can’t get the meaning!!
· Line 141: Between dt and will there should be a blank space!
· Line 161: “…may be large than…” should be “larger”.
· Line 212. “Reference source not found”. Please, revise.
· Lines 324 and 326. Please, leave a blank space before “Figure *”.
· Line 369. LHV stands for “lower heating value”, not “low heat value” (as written by the authors).
· Line 425. “…during analyzing is faster…” The sentence is incorrect. I can’t get the meaning.

Author Response
Dear reviewer,
Thank you very much for the excellent and professional revision of our manuscript. The manuscript is entitled as follows: “A zero-dimensional mixing controlled combustion model for real time performance simulation of marine two-stroke diesel engine”. We have carefully revised the manuscript according to your suggestion.
Our responses to the comments are listed below:
Comment: Line 92. The second equation in the manuscript is identified by (1), as the first equation. Consequently, the identification of all the equations is wrong. Authors should revise these identifications of the equations, as well as all the references to the equations appearing in the text.
Reply: Thanks. It had been revised.
Comment: Line 124. I don’t agree with this equation of the generalized excess air ratio, . If the equation is developed, you find that , where is the mass of pure product in the cylinder (i.e. stoichiometric burned gases), and is the un-combusted fuel. This is incorrect! What is correct is, but this is not what your equation is stating!!
Reply: The name of the generalized excess air ratio may be confusing. However, its meaning is clear. The generalized excess air ratio is defined by this paper. The mass of the un-combusted fuel can be obtained by eq.39 and eq.40. The mass of the injected fuel can be obtained by integrating eq.16. Then the mass of pure product can be calculated by the difference of the total mass and the un-combusted mass.
Comment: Equations in lines 129 and 130. There are several concerns about these equations. On the one hand, the R of a particular gas (Rz in this case) is obtained by dividing the universal R (Â) by the molecular weight of the gas: . The two equations given by the authors are not consistent with this relationship!! If we represent the evolution of Rz based on the authors’ equation (Rz authors) and the one coming from the equation I’ve just introduced (which depends on Mz), in both cases versus the generalized excess air ratio (), the trends are not the same (see figure below):
Reply: These two equations are NOT from theoretical analysis. Also, the component of the gas is changing follow the change of the general excess ratio. That means the gas isn’t any particular gas. It is a series of different gas.
We could find that the value of Rz/Mz is constant at different ak from the table below. So, we believe these two equation are right.
ak | 1.00 | 1.40 | 1.80 | 2.20 | 2.60 | 3.00 | 3.40 |
Rz | 286.69 | 286.81 | 286.87 | 286.91 | 286.94 | 286.96 | 286.97 |
Mz | 28.93 | 28.94 | 28.95 | 28.95 | 28.96 | 28.96 | 28.96 |
Rz/Mz | 9.91 | 9.91 | 9.91 | 9.91 | 9.91 | 9.91 | 9.91 |
Comment: On the other hand, I think it is not a good idea to use the constant g (acceleration of gravity) in the equation for Rz, since it makes no physical sense. If g has some relationship with Rz, it should be the reverse relationship: Rz is not proportional to g, but conversely proportional to g (or proportional to 1/g), because the mass is in the denominator of the equation I’ve indicated above. So, the equation given by the authors is misleading…
Reply: Yes, I think I miss a mistake. The equation has changed to equation below:
Comment: Line 177. In this graph, the authors show the impact of x on the combustion evolution. But unfortunately they don’t indicate which the value of the parameter that they finally use in their calculations is. I will latter comment a little bit more about this.
Reply: We use 1.4 for the engine we simulated. But the value should be different form engine to engine. We have no further information for other type of engines. So, we don’t think it is good to recommend the value.
Comment: Line 186. I assume that “continuous equation” should be “continuity equation”.
Reply: Sorry, we will modify.
Comment: Line 187. I’m a little bit confused with this equation. If Vn is the volume of the chamber volume of the nozzle, this is a constant magnitude. Why, then, dVn/dt appears in the equation?? This derivative should be 0!!
Reply: Yes, the chamber volume of the nozzle is constant if the needle valve lift needn’t to be considered. We revised the equation as below:
Comment: Line 203. In this equation, there should be a density multiplying in the term of the right of the equation. Otherwise, the units are not consistent in the equation.
Reply: Thanks a lot. The equation is revised as:
Comment: Line 220. In the equation proposed by the authors, the ambient density () affects the spray cone angle with an exponent 0.5 (a square root). This value seems to be too high. If you look at reference SAE 960034, from Naber and Siebers, they talk about an exponent of 0.19. Why have you used such a big exponent??
Reply: We followed the equation 17 in the references 35.
Comment: Line 269. I’m confused about this “reaction rate model”. If you are developing a combustion model which is mixing controlled, what’s the purpose of this “reaction rate model”?? Usually, in a MCC what you assume is that combustion takes place once mixing has taken place. This is much simpler than what you are doing (and more realistic). Any comment on that??
Reply: The reaction rate model is to adjust the combustion rate. The heat release rate model is based on the reaction rate model. The total reaction rate model can be found in eq.37, 38 and 41.
Also, the reaction rate can be used to adjust the ignition delay time, which in turn adjust the maximum pressure.
Comment: Line 318. Still one more confusing thing… If you have a mixing controlled combustion, which is a diffusive combustion, under these circumstances the combustion always takes place under near-stoichiometric conditions. Why, then, the authors introduce a discussion depending of the lambda of the combustion process?? This discussion is not relevant here, because combustion always takes place at lambda≈ 1!!
Reply: We assume the combustion only takes place on the surface of the injection cone. But the combustion rate or the reaction rate is directly adjusted by the lambda. Therefore, we have to introduce a model to calculate the lambda in the combustion volume. In fact, the combustion doesn’t always take place under near-stoichiometric conditions.
Furthermore, when the combustion takes place has no direct relationship with the lambda. It has direct relationship with the preparation process of the fuel gas, which can be found as eq.37. And the is determined by the reaction rate. When the integration of the reached a preset value, combustion can take place.
Comment: Figures 12 and 13. Looking at the experimental HRR, you should probably filter a little bit more the signal, to avoid this “ugly” oscillations in the curves.
Reply: Sorry.
Comment: Line 440. The authors state that no further explanation about the Simulink model presented in figure 14 is needed. But I have still some doubts of the role of some of the boxes: what is FOCR, VCU and ICU??
Reply: FOCR stands for Fuel Oil Common Rail. VCU stands for Valves Control Unit, which controls the exhaust valves. ICU stands for Injection Control Unit, which controls the fuel injection process. These three modules are less significant in this paper.
Comment: When showing the results for 2 different cases (i.e. two operating conditions), nothing is said about the set of model constants (for instance, x). Are they the same for both cases?? Or are they different?? This is very important to be known, since this would give more information about the prediction capability of the model.
Reply: We use the same model settings for the 2 different cases. The value of is 1.4.
Comment: Line 443. Finally, you present only results for Case 2, regarding the calculation speed. As far as I understand, you are simulating something that takes place in 60 seconds in 35.5 s (in average), isn’t it? This is not very clear in the text. If you say that “the simulation time was set to 60 seconds”, it is not very clear if this is the time for the physical process, or if this is the time for the simulation. Please, clarify. And the other thing is that it would be nice if you show the same information but for Case 1 (even if it is probably less exigent –because of the lower engine speed-), since this will “enrich” the information given in the paper regarding suitability of the model for real time calculations.
Reply: Thank you very much.
Additionally, we have checked English grammar, spelling and sentence structure carefully in the revised manuscript.
Thank you and best regards.
Yours sincerely,
Yongming Feng

Round 2
Reviewer 1 Report
Thank you for reviewing the paper. Now it is eligible for publication.